# M3DOnline: Foundation-Prior Guided Monocular 3D Motion Learning for Autonomous Driving in Novel Scenes

## Abstract

We propose M3DOnline, a learning framework for normalized scene flow (NSF). NSF represents the dense 3D motion of pixels between two frames and plays a critical role in various monocular 3D vision tasks. Existing self-supervised NSF methods heavily rely on strong visual cues, which limits their performance on non-Lambertian surfaces and around motion boundaries. Our key insight is to leverage useful priors from foundation models to overcome the inherent limitations of texture-based matching in traditional self-supervised methods. Specifically, we design a pseudo-label generation pipeline using semantic and depth foundation models. Based on rigid motion assumptions, we divide real-world scenes into semantic segments and generate per-segment 3D motion pseudo-labels. To handle inevitable non-rigid regions and reduce the impact of inaccurate predictions from foundation models, we introduce a loss-based adaptive learning strategy, which filters out obvious non-rigid areas and dynamically adjusts the learning weight and region based on label quality. Experiments show that M3DOnline significantly improves motion boundary estimation and the handling of reflective and transparent surfaces. This demonstrates the advantage of integrating foundation model priors into self-supervised scene flow learning. Code will be available.

## 1 Introduction

Monocular 3D motion estimation, also known as normalized scene flow (NSF), aims to estimate dense optical flow and motion-in-depth (MID) between two consecutive frames Nefs et al. (2010). It is a key technique for enabling computers to understand the physical world. With NSF being increasingly applied in autonomous driving, scene reconstruction, and action recognition Yuan et al. (2023); Menze et al. (2018a); Zhang et al. (2024); Badki et al. (2021); Yang et al. (2012), researchers are gradually considering its capacity for continual learning in real-world scenarios.

At present, state-of-the-art (SOTA) methods for NSF Yang (2020); Ling et al. (2022; 2023); Badki et al. (2021); Yang & Ramanan (2021); Ling et al. (2024c) have constructed deep network structures under a supervised learning framework, achieving excellent performance on benchmark datasets. However, obtaining dense ground truth labels in new scenarios is difficult Menze et al. (2018a; 2015), as it requires synchronous data from multiple cross-modal sensors and extensive manual annotations. This significantly limits the scalability of SOTA methods to unseen scenarios Yuan et al. (2022).

A promising solution is self-supervised learning, which indirectly supervises models through photometric loss, achieving label-free learning of NSF in new environments Hur & Roth (2020; 2021); Stone et al. (2021); Bendig et al. (2022). However, many challenges remain. Firstly, self-supervised learning relies on photometric loss based on color and texture, lacks high-level contextual information guidance, and struggles with non-Lambertian[1] surface Tagare & Defigueiredo (1993) and boundary-blurring problems Stone et al. (2021); Yuan et al. (2024). Secondly, indirect supervision forms makes it difficult to train complex models. Currently, most loss-driven self-supervised methods Hur & Roth (2021; 2020) can only train customized lightweight networks, limiting their capacity to learn sufficient knowledge for generalization Stone et al. (2021); Jonschkowski et al. (2020).

---

[1]The brightness of a non-Lambertian surface changes with the observer's viewing angle.

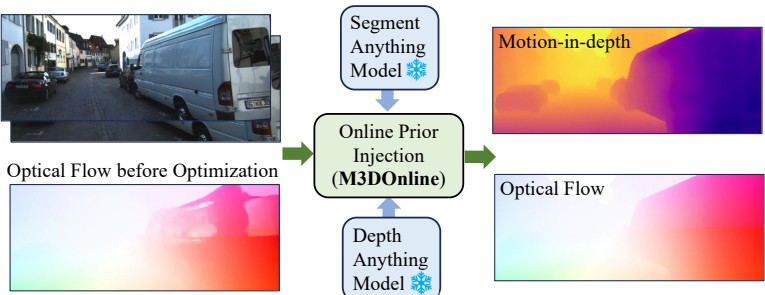

Figure 1: **Our M3DOnline.** M3DOnline leverages semantic and depth priors from the Segment and Depth Anything models, enhancing the 3D motion model's capability in estimating motion boundaries and reflective surfaces in novel environments.

Fortunately, the recently popular pre-trained visual models Segment Anything (SAM) Kirillov et al. (2023) and Depth Anything (DAM) Yang et al. (2024) have the potential to address these issues. For non-Lambertian surface problems, DAM can reliably extract geometric structure information to guide the NSF estimation. For texture dependency issues, SAM provides high-level contextual information and accurate target-level edge constraints Yuan et al. (2024). Therefore, we proposes a data-driven self-supervised NSF learning framework, M3DOnline, to infuse semantic and 3D structural knowledge into NSF through pseudo-label training, aiming to overcome the inherent limitations of previous loss-driven self-supervised schemes.

To inject prior knowledge into the NSF model, M3DOnline first generates pseudo labels from input priors based on rigid optimization principles. This approach is motivated by the observation that most moving objects in everyday scenes approximately follow rigid motion, as demonstrated in many prior worksTeed & Deng (2021); Liu et al. (2022). To deal with inevitable non-rigid motions and potential errors introduced by imperfect foundation models, M3DOnline further incorporates a set of label reliability evaluations and a loss-based dynamic weighting mechanism to enable robust knowledge integration.

Experimentally, in the time-to-collision task, M3DOnline outperforms existing self-supervised methods and achieves accuracy comparable to supervised approaches. On the core metric SF-all of the KITTI 15 scene flow public benchmark, M3DOnline achieved an outlier rate of 8.46%, surpassing some supervised methods Badki et al. (2021). As shown in the Fig. 1, M3DOnline effectively mitigates the inherent limitations of previous self-supervised approaches, enabling NSF to learn clearer and more accurate 3D motion in novel scenarios.

Our key contributions can be summarized as follows:

1. To the best of our knowledge, we are the first to effectively integrate SAM and DAM models with self-supervised NSF, enabling 3D motion learning from real-world road videos without ground-truth labels.

2. An effective dynamic weight-based training strategy, which can flexibly utilize pseudo labels of different qualities, avoiding label errors affecting model training.

3. In multiple downstream 3D task evaluations, M3DOnline achieved performance comparable to supervised methods without using scene flow labels.

## 2 RELATE WORKS

### 2.1 LOSS-DRIVEN SELF-SUPERVISION

The previous self-supervised NSF methods Hur & Roth (2021; 2020); Yang (2020) optimizes motion estimation using photometric loss combined with a series of regularization losses. Specifically, photometric loss is widely used for self-supervised optical flow and depth estimation, while spatial endpoint loss indirectly improves motion-in-depth. However, self-supervised methods Horn & Schunck (1981); Zhang et al. (2021); Janai et al. (2018); Liu et al. (2021); Hur & Roth (2019) based on photometric loss inherently suffer from a key limitation: they rely on significant visual motion

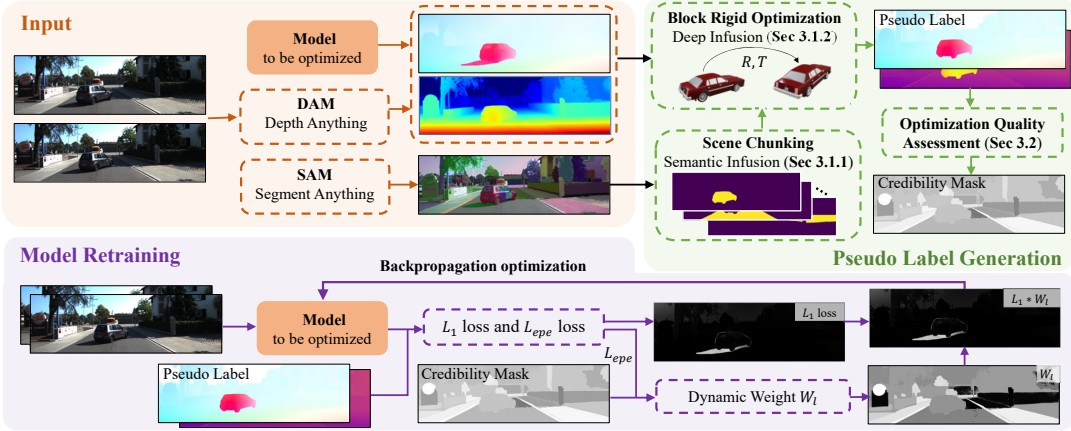

Figure 2: **Overview of M3DOnline.** pseudo-labels with confidence masks are derived from structural and semantic priors of foundation models. During retraining, label weights are dynamically adjusted based on the current loss for robust knowledge injection.

rather than the physical motion of objects, leading to challenges in handling motion boundaries and non-Lambertian surfaces. In contrast, our M3DOnline optimizes the 3D motion field by leveraging the motion of physically rigid scene structures, enabling better handling of non-Lambertian surfaces in driving scenarios and improving robustness at motion boundaries.

## 2.2 PRE-TRAINED MODEL AND MOTION ESTIMATION

Due to its strong generalization ability and competitive performance, SAM Kirillov et al. (2023) has been preliminarily applied as a segmentation model in optical flow estimation. Among these applications, SAMFlow Zhou et al. (2024) leverages SAM-generated features as contextual guidance, significantly enhancing the optical flow model's object-level perception. Similarly, UnSAM Yuan et al. (2024) segments scenes using SAM, refines scene blocks via homography transformation, and incorporates SAM's semantic segmentation into the optical flow network, expanding its adaptability in optical flow estimation. However, these methods require SAM during inference, introducing additional computational overhead.

In contrast, M3DOnline utilizes SAM offline, eliminating runtime dependencies and maintaining efficiency across different models. Additionally, M3DOnline integrates the latest Depth Anything model (DAM) Yang et al. (2024) to refine the scene motion field based on rigid motion principles. This helps to address the inherent dependence of previous self-supervised methods on texture.

## 3 METHOD

This section introduces how to inject depth and semantic priors into existing NSF models. It is specifically divided into two stages. The first stage is the generation of NSF pseudo labels, which aims to transform the scene structure and semantic knowledge into pseudo labels. The second stage is to retrain the original model using these pseudo labels to achieve knowledge injection. As shown in Fig. 2, in this stage, the main approach is to prevent non-rigid interference and prior errors by evaluating label credibility and dynamic loss weighting mechanisms. Unlike previous work Kirillov et al. (2023); Yuan et al. (2024), we used an offline model usage scheme (the Anything model does not participate in inference), which significantly improves the efficiency of method training and inference, broadening the applicability of our solution to the model.

### 3.1 PSEUDO LABEL GENERATION

Numerous prior worksMenze et al. (2018b); Yang & Ramanan (2021); Liu et al. (2022) have demonstrated that the rigidity assumption can explain most motion processes observed in the real world. Even seemingly deformable objects, such as the human body, can be regarded as a composition

of rigid blocks connected by joints when sufficiently subdivided. Based on this solid assumption, we extract pseudo-labels from priors. First, we segment the scene into sufficiently small regions using SAM, aiming to decompose the image into appropriate rigid units. Second, we individually perform rigid optimization on each segment region to compute their corresponding optical flow and motion-in-depth labels.

### 3.1.1 SCENE CHUNKING AND OPTIMIZATION ORDER

Decomposing the scene into rigid blocks is crucial, as ensuring that the pixels within each block mask $M_o$ belong to the same rigid body is a prerequisite for rigid optimization. In previous scene flow works Yang & Ramanan (2021); Liu et al. (2022), overall rigid background optimization was one of the most commonly used techniques. It can diffuse prior depth information to the background through a small amount of correct optical flow. However, accurately estimating the depth of the overall background remains a challenge for DAM. To address this, we leverage SAM's "everything" mode to divide the scene into fragments, thereby avoiding erroneous depth interference while providing accurate target-level edges for pseudo labels.

**Optimization Order:** The mask input used for optimization is divided into two categories: the mask set $M_s$ provided by SAM and the overall background mask $M_b$. When optimizing, we first optimize $M_b$ followed by sequential optimization of the mask blocks in $M_s$, if overlapping areas are encountered, scene blocks with higher optimization quality are prioritized. Moreover, blocks with insufficient optimization quality are directly discarded to save computing resources.

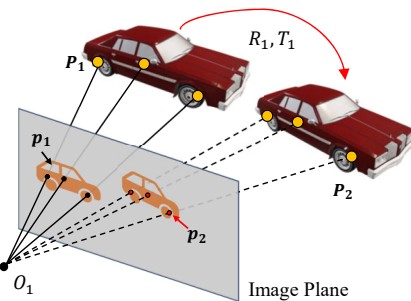

Figure 3: **An Example of Rigid Motion.** The red car is a rigid body that moves from $P_1$ to $P_2$ in two consecutive frames. The corresponding projection on the image plane moves from $p_1$ to $p_2$. After knowing $p_1$, $p_2$, and $P_1$, the specific motion trajectory $(R_1, T_1)$ of the car between two frames can be obtained by solving the PnP problem.

### 3.1.2 LABEL GENERATION BASED ON RIGID OPTIMIZATION

The core idea of rigid optimization is to calculate the motion trend of the rigid body through a small amount of correct optical flow, then combine it with depth to determine the overall optical flow and motion-in-depth field. There are two benefits to doing this. Firstly, we can optimize other parts based on a small amount of correct estimation. Secondly, we can estimate areas difficult for algorithms to handle, such as occlusion and lack of texture. Next, we introduce the specific steps of the rigid optimization module.

**Input:** RGB images of two consecutive frames $I_1, I_2$; Estimation of optical flow field $f$; The depth of the first frame estimated by the DAM $Z_1$; Mask of the block to be optimized $M_o$.

**Rigid Optimization:** Taking Fig. 3 as an example. The pixel point set corresponding to mask $M_o$ (the orange car on the left) is $p_1 = \{p \mid M_o(p) = 1, p = (u, v)\}$. Assuming the camera is stationary relative to the world coordinate system, the car moves from $p_1$ to $p_2$ between two frames. After calculating the 3D point set $P_1 = \{P \mid P = (X, Y, Z)\}$ corresponding to $p_1$, we can use the PnP Hesch & Roumeliotis (2011); Lepetit et al. (2009) algorithm to get the car's motion $(R_1, T_1)$ relative to the camera:

$$P_1 = Z_1(p_1)K^{-1}p_1 \tag{1}$$

$$p_2 = p_1 + f(p_1) \tag{2}$$

$$(R_1, T_1) = \text{SolvePnP}(\boldsymbol{P_1}, \boldsymbol{p_2}, K) \tag{3}$$

where $K$ is the camera's internal parameter, $R_1 \in \mathbf{SO(3)}$ represents car's rotation, $T_1 \in \mathbb{R}^3$ represents car's translation. Next, we recalculate the optical flow field $\boldsymbol{f'(p_1)}$ and motion-in-depth field $\boldsymbol{\tau'(p_1)}$ corresponding to $M_o$ based on the car motion $(R_1, T_1)$:

$$\boldsymbol{P_{2'}} = R_1 \boldsymbol{P_1} + T_1 \tag{4}$$

$$\boldsymbol{p_{2'}} = \frac{K\boldsymbol{P_{2'}}}{Z_{2'}(\boldsymbol{p_{2'}})} \tag{5}$$

$$\boldsymbol{f'(p_1)} = \boldsymbol{p_{2'}} - \boldsymbol{p_1} \tag{6}$$

$$\boldsymbol{\tau'(p_1)} = Z_{2'}(\boldsymbol{p_{2'}})/Z_1(\boldsymbol{p_1}) \tag{7}$$

where $\boldsymbol{f'(p_1)}$ and $\boldsymbol{\tau'(p_1)}$ is the NSF after rigid optimization, $Z_{2'}(\boldsymbol{p_{2'}})$ is the value of $Z$ in the set of 3D points $\boldsymbol{P_{2'}}$.

## 3.2 Optimization Quality Assessment

In this section, we propose a comprehensive evaluation scheme to assess the quality of rigid optimization. Assessment is a combination of the structural similarity (SSIM) Wang et al. (2004) metric and corresponding credibility $C$. SSIM directly evaluates the correctness of the generated pseudo-optical flow labels, while credibility prevents SSIM failure caused by low texture, occlusion, and overexposure.

**Base Assessment:** We warp the region corresponding to mask $M_o$ from $I_2$ to $I_1$ based on the optical flow pseudo label $\boldsymbol{f'}$ to obtain $I_{1'}(M_o)$. Next, use the structural similarity function to calculate the similarity between the original and warp images: $S_{base} = \text{SSIM}(I_1(M_o), I_{1'}(M_o))$. The larger the $S_{base}$, the higher the optimization quality.

**Effective Evaluation of The Area:** SSIM is a texture-based quality metric, which may not work reliably in textureless or occluded regions. To address this, we computed the image gradient and selected regions with prominent gradient responses. Moreover, we excluded occluded regions based on a consistency check Wang et al. (2018). The final effective mask $M_{used}$ for evaluation is defined as follows:

$$M_{used} = M_o M_{noc} M_g \tag{8}$$

where $M_{used}$ is the effective region for $M_o$, $M_{noc}$ is the non-occluded mask for bidirectional optical flow consistency calculation, and $M_g$ is the region with a significant Sobel gradient response.

**Assess Credibility:** In order to prevent various extreme situations and better measure the quality of optimization, we also propose the following credibility coefficients from the perspectives of entropy value $E$, average brightness $L$, and effective matching area proportion $P$ of the matching area:

$$E(M_o) = \tanh((\mathbf{Entropy}(I_1(M_{used})) - k_1)^3) \tag{9}$$

$$L(M_o) = \tanh((\mathbf{Mean}(I_1(M_{used})) - k_2)^{0.1}) \tag{10}$$

$$P(M_o) = \frac{\mathbf{Sum}(M_{used})}{\mathbf{Sum}(M_o)} \tag{11}$$

$$C(M_o) = k_3(E(M_o) + L(M_o) + P(M_o)) \tag{12}$$

where $C(M_o)$ is the credibility coefficient corresponding to mask $M_o$, **Entropy** is the calculation function of image entropy, the value ranges of $E$ and $L$ are standardized between -1 and 1, the value ranges of $P$ is standardized between 0 and 1, $k_1$ and $k_2$ are truncation coefficients generally set to 4 and 0.1, $k_3$ is used to adjust the value range of $C(M_o)$, which is generally set to 0.5. The larger the value of $C$, the higher the credibility of the mask $M_o$.

Finally, the pseudo label quality assessment metric $S$ is defined as follows:

$$S(M_o) = \text{Mean}(S_{base} M_{used} C(M_o)) \tag{13}$$

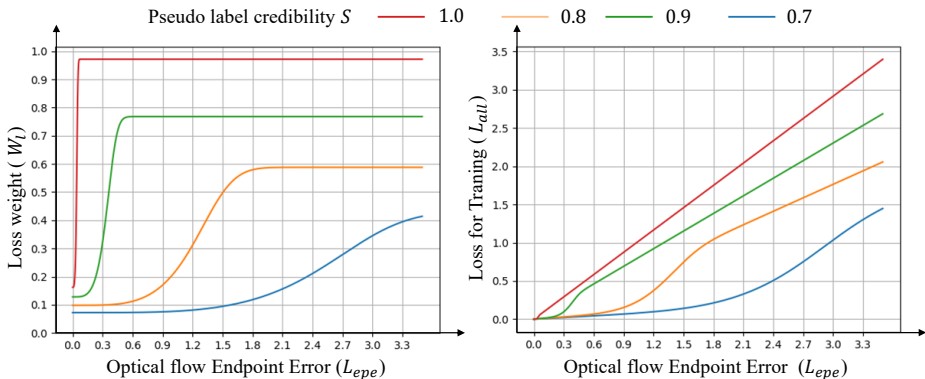

Figure 4: **Visualized Dynamic Loss Weights.**

### 3.3 KNOWLEDGE INJECTION

We propose a dynamic weight-based training pipeline to achieve robust knowledge injection, the core idea of which is to assign different effective training ranges and weights to pseudo labels with different quality levels. Simply put, for low-quality labels, there is only a significant learning weight when the estimated results differ significantly from the label values. In contrast, for high-quality labels, there is always a higher learning weight. The advantage of doing so is that it can fully utilize low-quality pseudo labels. Although low-quality pseudo labels may have some errors compared to the true values, they are generally correct in direction and can be used for rough learning.

Specifically, we implement this idea through a dynamic loss coefficient $W_l$ based on quality metric $S$ and optical flow endpoint error $L_{epe}$:

$$L_{epe} = \sqrt{(f_1 - f_1')^2 + (f_2 - f_2')^2} \tag{14}$$

$$W_l(L_{epe}, S) = S^2(\tanh(mL_{epe}^4) + 0.2) \tag{15}$$

where $(f_1, f_2)$ is the optical flow estimation result, $(f_1', f_2')$ is the optical flow pseudo label, and we limit the range of $S$ to between 0 and 1. As shown in Fig. 4 $W_l$ is a function about $L_{epe}$ and $S$, $m = 10^{k_6 S^5 - 3}$ is the adjustment coefficient for $W_l$. The larger the $k_6$, the more inclined $W_l$ is to believe in the pseudo label.

The final loss $L_{all}$ based on $W_l$ can be defined as:

$$L_1 = \|f_1 - f_1'\|_1 + \|f_2 - f_2'\|_1 + \|\boldsymbol{\tau} - \boldsymbol{\tau}'\|_1 \tag{16}$$

$$L_{all} = W_l(L_{epe}, S)L_1 \tag{17}$$

## 4 EXPERIMENT

**Datasets:** During training, we used 4198 image pairs from the KITTI 2015 Menze et al. (2015) multi-frame training set (Km), 3991 image pairs from the KITTI 2015 multi-frame testing set (Kme), 43560 image pairs from the KITTI raw (Kr), 2082 image pairs from Sintel training set (S-train), and 15800 image pairs generated by SAG (SAG15) Ling et al. (2024b). In the evaluation, we used 200 image pairs from the KITTI 2015 training set (K15), as well as KITTI 2015 (K15-test), Sintel testing set (S-test), and KITTI 2012 (K12-test) public testing platforms.

In the TTC experiment, to maintain the same evaluation criteria as previous work Yang (2020); Ling et al. (2024a), we took one out of every five images from K15 to obtain a subset with a size of 40 (K40).

**Training Details:** We chose the self-supervised version of ScaleFlow++ from SAG Ling et al. (2024b) as the baseline model for generating pseudo labels. During training, we first generated pseudo-labeled datasets **Kme***,**Kr***,and **Km***. In the NSF, scene flow, and TTC experiments, we fine-tuned the

pre-trained ScaleFlow++ model using a mixed dataset with **Kr***, **Km***, and SAG15 Ling et al. (2024b) (Batch=6, iteration=80K, LR = 0.0001). In the ablation experiment, we trained from scratch using **Kme*** mixed SAG15 (Batch=6, iteration=180K, LR=0.0025).

Table 1: **Time-to-collision Evaluation on K40. Top** is the **supervision** method, **bottom** is the **self-supervision** method.

| Method | Err-1s | Err-2s | Err-5s | Time/s |
|---|---|---|---|---|
| TPCV Ling et al. (2023) | 1.14 | 1.65 | 1.57 | 0.2 |
| Expansion Yang (2020) | 1.86 | 1.87 | 2.45 | 0.20 |
| OSF Menze et al. (2018b) | 1.79 | 2.93 | 4.03 | 3000 |
| Binary TTC Badki et al. (2021) | 2.40 | 3.16 | 3.25 | 0.02 |
| SMMSF Hur & Roth (2021) | 8.31 | 8.46 | 10.60 | **0.06** |
| SAG Ling et al. (2024b) | 3.59 | 3.78 | 3.54 | 0.2 |
| M3DOnline (ours) | **2.10** | **2.84** | **2.71** | 0.2 |

### 4.1 TIME-TO-COLLISION

Time-to-collision reveals the time required for an object to make contact with the camera plane, which is crucial for autonomous driving obstacle avoidance and path-planning tasks. TTC can be indirectly calculated based on MID $\tau$:

$$TTC = \frac{Z}{Z - Z'}T = \frac{T}{1 - \frac{Z'}{Z}} = \frac{T}{1 - \tau} \qquad (18)$$

where $T$ is the sampling interval of the camera (in KITTI $T = 0.1s$). Referring to the previous evaluation criteria Yang (2020), we consider TTC as a binary classification problem. For each pixel, we determine whether its TTC is less than $\{1s, 2s, 5s\}$. Only the points moving towards the camera are evaluated.

As shown in the Tab. 1, M3DOnline outperformed all self-supervised methods. Moreover, even compared to supervised methods, M3DOnline demonstrates strong competitiveness, surpassing the supervised Binary TTC Badki et al. (2021) across all time intervals. This result suggests that the M3DOnline generalization framework has the potential to facilitate data-efficient TTC.

### 4.2 SCENE FLOW BENCHMARK

Scene flow reveals the 3D motion state of objects in space, playing a vital role in behavior prediction and path planning for autonomous driving. It consists of optical flow and corresponding pixels' depth variation between two frames. To ensure consistency with state-of-the-art evaluation criteria, we

Table 2: **State-of-the-art Published Monocular Methods on KITTI Scene Flow Benchmark.** D1, D2, Fl, and SF is the percentage of disparity, optical flow and scene flow outliers. -bg,-fg and -all represent the percentage of outliers averaged only over background regions, foreground regions and overall ground truth pixels. The best among the same group are bolded. **Top** is the **supervision** method, **bottom** is the **self-supervision** method.

| Method | D1-all | D2-all | Fl-bg | Fl-fg | Fl-all | SF-bg | SF-fg | SF-all |
|---|---|---|---|---|---|---|---|---|
| Optical expansion Yang (2020) | 1.81 | 4.25 | 5.83 | 8.66 | 6.3 | 7.06 | 13.44 | 8.12 |
| Binary TTC Badki et al. (2021) | 1.81 | 4.76 | 5.84 | 8.67 | 6.31 | 7.45 | 13.74 | 8.5 |
| TPCV Ling et al. (2023) | 1.81 | 3.18 | 4.53 | 5.52 | 4.69 | 5.34 | 10.60 | 6.21 |
| Scale-flow Ling et al. (2022) | 1.81 | 2.55 | 5.24 | 5.71 | 5.32 | 6.06 | 11.32 | 6.94 |
| SMMSF Hur & Roth (2021) | 22.71 | 26.51 | 12.41 | 18.20 | 13.37 | 31.18 | 42.68 | 33.09 |
| ADFactory Ling et al. (2024a) | 1.81 | 7.71 | 11.28 | 10.66 | 11.18 | 13.19 | 16.18 | 13.68 |
| SAG Ling et al. (2024b) | 1.81 | 6.22 | 9.78 | **9.50** | 9.73 | 11.83 | **14.69** | 12.31 |
| M3DOnline (ours) | 1.81 | **4.11** | **5.68** | 11.06 | **6.58** | **6.83** | 16.63 | **8.46** |

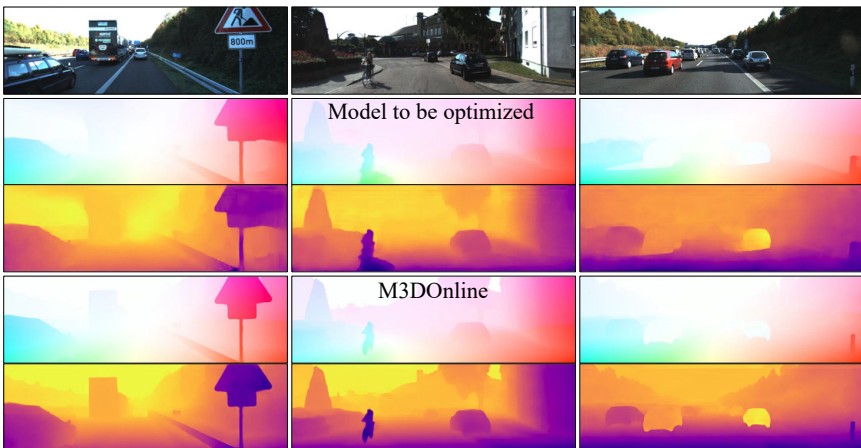

Figure 5: **Visual Performance Comparison in Typical Driving Scenarios.** M3DOnline significantly improved the 3D motion estimation results of motion edges, non-Lambertian surfaces, and rider areas (red box).

utilized GANet Zhang et al. (2019) to estimate the depth $D1$ of the first frame and computed the depth variation using MID $\tau$, where the depth of the second frame is given by $D2 = \tau \cdot D1$.

As shown in Tab. 2, our proposed M3DOnline outperforms self-supervised NSF in most evaluation metrics. Moreover, M3DOnline remains competitive even when compared to supervised methods (8.46 v.s. 8.5).

Compared with the SAG-pretrained ScaleFlow++ Ling et al. (2024b), M3DOnline exhibits significant improvements in SF-all (8.46 v.s. 12.31), and demonstrates superior background prediction (6.83 v.s. 11.83), highlighting its effectiveness in optimizing motion edges. However, the optimized M3DOnline exhibits slightly lower foreground performance than SAG, primarily due to two factors. First, non-Lambertian surfaces are largely excluded from the evaluation, preventing the improvements made by M3DOnline from being fully reflected. Second, the DAM model in M3DOnline is not specifically tailored for driving scenarios, leading to inevitable prediction errors. We have demonstrated in Sec. 4.4 that depth accuracy is positively correlated with the overall performance of M3DOnline.

Table 3: **Self-supervised Optical Flow Evaluation. mf\*** means that a large model SAM is used to participate in reasoning.

|  | KITTI15-test | | KITTI12-test | Sintel-test | |
| --- | --- | --- | --- | --- | --- |
|  | Fl-all | all-noc | $L_{epe}$ | Fl-all(clean) | Fl-all(final) |
| UnSAM+**mf\***Yuan et al. (2024) | 7.83 | 5.67 | 1.40 | 3.93 | 5.20 |
| UFlow Jonschkowski et al. (2020) | 11.13 | 8.41 | 1.90 | 5.21 | 6.50 |
| SMURF Stone et al. (2021) | 6.83 | 5.26 | 1.40 | 3.15 | **4.18** |
| UPFlow Luo et al. (2021) | 9.38 | - | - | 4.68 | 5.32 |
| DDFlow Liu et al. (2019) | 14.29 | 9.55 | 3.00 | 6.18 | 7.40 |
| M3DOnline (ours) | **6.58** | **4.71** | **1.40** | **2.10** | 4.43 |

### 4.3 SELF-SUPERVISED OPTICAL FLOW

We also compared M3DOnline with self-supervised methods focused on optical flow and evaluated them on the KITTI and Sintel datasets. To evaluate Sintel, we additionally generated a pseudo-labeled dataset on S-train and used the same training process as KITTI.

As shown in Tab. 3, in KITTI's optical flow benchmark, M3DOnline consistently outperforms other self-supervised methods. However, in the Sintel benchmark, the advantage of M3DOnline is less pronounced, primarily due to the presence of numerous nonrigid objects, which pose challenges

for generating reliable pseudo-labels. Nevertheless, M3DOnline can still handle mildly nonrigid objects effectively in real-world scenarios, as illustrated in Fig. 5. After being refined by M3DOnline, ScaleFlow++ accurately estimated the rider's motion state.

## 4.4 ABLATION STUDY

This section designs a series of ablation experiments to verify the effectiveness of key modules in M3DOnline. We generated a pseudo-labeled dataset **Kme\*** on the KITTI test set Kme and tested it on the training set K15, which contained groundtruth.

Table 4: **Ablation Study.** $MID_{err}$ is the motion-in-depth evaluation metric, refer to Yang (2020) for details.

| ID | DAM | SAM | Dynamic Weights | Assessment S | $L_{epe}$ | Fl-all | $MID_{err}$ |
|----|-----|-----|-----------------|--------------|-----------|--------|-------------|
| A | large | ✓ | ✓ | ✓ | 2.23 | 7.3 | 90.08 |
| B | large | ✓ | ✓ | ✗ | 2.51 | 9.31 | 98.12 |
| C | large | ✓ | ✗ | ✓ | 2.46 | 8.35 | 96.21 |
| D | large | ✗ | ✗ | ✗ | 10.14 | 30.05 | 228.41 |
| E | ✗ | ✗ | ✗ | ✗ | 3.72 | 12.15 | 180.56 |
| F | base | ✓ | ✓ | ✓ | 2.4 | 8.3 | 96.27 |
| G | small | ✓ | ✓ | ✓ | 1.75 | 11.52 | 107.48 |

**Assessment $S$:** We remove the low-quality label filtering based on the assessment metric $S$ in **B**. By comparing **A** and **B**, we observe a drop in all evaluation metrics, indicating that $S$ effectively distinguishes and filters out noisy labels caused by non-rigid motion or prior errors.

**Dynamic Weight:** We removed the dynamic learning weights from the retraining stage. As shown in Tab. 4 **A** and **C**, there is a decrease in performance in all indicators. This indicates that the pseudo labels generated directly by SAM and DAM contain interference that affects training, and it is necessary to identify them before use.

**SAM Segmentation:** To verify the rationality of the SAM block optimization strategy, we further ablated it based on the **C** setting. In this case, the pseudo-label generation pipeline relies solely on global background rigid optimization (similar to Liu et al. (2022)). As shown in Tab. 4 **D**, the performance of all metrics significantly declines (2.23 v.s. 10.14), indicating that the global depth error rate provided by DAM is high, leading to pseudo-labels with substantial noise. Local optimization after SAM-based partitioning can effectively mitigate this issue.

**Compare to Baseline:** We also compared M3DOnline with the baseline model trained solely on the SAG15 dataset. As shown in Tab. 4 **A** and **E**, incorporating all the proposed modules, led to an overall performance improvement of nearly 50%. This demonstrates that the proposed modules, when used in combination, can significantly enhance the baseline model's performance in new scenarios.

**Depth Accuracy:** We further investigated the impact of different levels of depth precision on pseudo-label generation. In previous experiments, the "**large**" version of DAM weights was used by default. In this ablation study, we additionally tested the "**small**" and "**base**" versions of the weights. As shown in Tab. 4 **F** and **G**, using suboptimal depth estimation led to a slight decline in model performance, demonstrating that the performance of M3DOnline is positively correlated with the depth estimation capability of DAM.

## 5 CONCLUSION

We propose M3DOnline, a framework for self-supervised learning of normalized scene flow (NSF) models. By incorporating prior knowledge of scene structure and semantics, M3DOnline significantly enhances the performance of NSF methods, particularly on non-Lambertian surfaces and moving edges. Furthermore, compared to previous self-supervised approaches, it produces clearer and more accurate NSF results .We believe that M3DOnline marks an important step toward label-free 3D motion estimation.

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
