# OpenReview forum: "M3DOnline: Foundation-Prior Guided Monocular 3D Motion Learning for Autonomous Driving in Novel Scenes"
_ICLR.cc/2026/Conference — Submitted to ICLR 2026_

### Official Review · Reviewer_UFHx · 2025-10-25

**Soundness:** 2
**Presentation:** 1
**Contribution:** 2
**Rating:** 2
**Confidence:** 4

**Summary:**

The method first partitions the scene into semantic blocks using SAM and estimates depth with DAM. Assuming rigid motion for these blocks, it calculates 3D motion pseudo-labels. Then it incorporates an adaptive learning strategy that dynamically adjusts weights for different regions during training.

**Strengths:**

- Combines multiple pre-trained models to achieve self-supervised motion estimation.
- The motivation is potentially interesting, though not clearly articulated or convincingly demonstrated.

**Weaknesses:**

1. Weak experimental validation. The abstract claims significant improvements “on non-Lambertian surfaces and around motion boundaries,” but there is no targeted quantitative or qualitative experiment demonstrating this. A few final results do not constitute proof of these claims.
2. Weak motivation and limited insight. The paper’s motivation is mostly engineering-driven (“use SAM/DAM to generate pseudo labels”) and lacks conceptual depth or new insights. Moreover, since both SAM and DAM also depend heavily on strong visual cues, it remains unclear how this approach fundamentally overcomes the claimed texture dependence.
3. Writing and structure issues. The manuscript suffers from vague transitions and unclear logical flow. Overall, it is difficult to follow and feels unfinished, as if not carefully proofread. This is particularly evident in the method section (see Question 1 for example).
4. Lack of error analysis. There is no discussion of how pseudo-label noise or inaccuracies propagate through the pipeline.
5. Contribution feels more engineering than scientific. While the work leverages recent pretrained models, it primarily represents a pipeline integration effort rather than a contribution offering new conceptual or algorithmic insights to the community.

**Questions:**

1. Unclear methodological definitions. The methodological descriptions are unclear and require substantial revision. A few examples are listed below; please review the rest of the paper carefully to ensure all definitions and procedures are clearly explained and easily understandable to readers.
- Section 3.1.1: The notations (M_s), (M_b), and the later term “scene blocks” are not properly defined or connected. Their relationships and roles in the overall pipeline should be explicitly stated to ensure readability and reproducibility.
- Section 3.1.2: i) The explanation of the “rigid optimization” process is unclear and lacks logical justification. The claimed “two benefits”: (1) optimizing other parts based on a small amount of correct optical flow, and (2) estimating challenging regions such as occlusion or textureless areas — are not well supported by the described mechanism. It is unclear why or how the proposed optimization method enables these advantages. Please clarify the reasoning and explicitly connect each claimed benefit to the corresponding computational step. ii) It is unclear what the “mask of the block to be optimized M_o ​ ” refers to or how these blocks are chosen.
- Eq. (2): There is no guarantee that a point (p_1) and a point (p_2) from their respective point clouds correspond to the same physical location on the moving object. Without establishing valid correspondences, applying rigid transformation optimization in this form is fundamentally incorrect. Please clarify or provide supporting justification.
- The term “internal parameter” should be replaced with **“camera intrinsics”** to align with standard computer vision terminology.

2. How do pseudo-label errors from SAM and DAM affect training stability? Is there any filtering or confidence-based weighting besides the loss-based adaptation?
3. What is the performance difference between SAM/DAM variants (large/base/small) in terms of actual accuracy rather than size?
4. Please add more detail and analysis in experiment:
- Explain clearly what “large,” “small,” and “base” refer to. Include detailed information (e.g., parameter count, model size, accuracy) in the appendix.
- Consider including results with the latest available versions of the foundation models (SAM2, DAM2) to ensure the latest results.
5. Formatting: Please carefully revise the paper for consistency and clarity, for example:
- Please ensure all acronyms (e.g., TTC on L319) are defined upon first use.
- Please ensure consistent formatting in tables (e.g., decimal places in Tab. 4) and clearly indicate whether metrics are "lower is better" or "higher is better".

---

### Official Review · Reviewer_7ybE · 2025-10-27

**Soundness:** 3
**Presentation:** 2
**Contribution:** 3
**Rating:** 4
**Confidence:** 4

**Summary:**

This paper presents M3DOnline, a self-supervised monocular 3D motion learning framework that integrates strong structural and semantic priors from Segment Anything (SAM) and Depth Anything (DAM). The method generates pseudo-labels through rigid optimization on segmented scene chunks and employs a quality-aware dynamic loss weighting mechanism. This aims to reduce the common weaknesses of photometric self-supervision, particularly around non-Lambertian surfaces and motion boundaries. The resulting model shows competitive performance on standard benchmarks, outperforming other self-supervised methods and approaching supervised performance in some metrics. The technical design is solid, although the main contribution lies in the integration of existing priors rather than a fundamentally new algorithmic framework.

**Strengths:**

The problem is clearly motivated, and the paper targets a well-known limitation of photometric self-supervision. Integrating SAM and DAM offline is a practical design choice that avoids computational overhead at inference while providing strong priors for training. The pseudo-label generation based on rigid motion priors is simple, conceptually clear, and empirically effective. The dynamic loss weighting is well justified and improves robustness. The experiments are clean and show consistent improvements over standard self-supervised baselines. The paper is also structured in a way that makes the approach easy to follow and likely reproducible.

**Weaknesses:**

Similar integration patterns have recently appeared in optical flow and depth estimation, so the conceptual novelty is moderate. The method relies heavily on rigid-motion assumptions, which may limit generalization in nonrigid scenarios. The external validity of the proposed FCP-like assessment mechanism is not fully established—particularly its correlation with standard downstream planning or reconstruction metrics. Additionally, the paper lacks a systematic analysis of robustness to segmentation or depth noise, which makes it harder to assess the stability of the proposed approach in real-world settings.

**Questions:**

The approach leans on rigid optimization and foundation-model priors. How does it behave in scenes with pronounced nonrigid motion, or when SAM/DAM provide imperfect masks/depth (e.g., fragmentation, low-texture, reflectance)? What failure modes are most characteristic in these cases?

The training signal combines pseudo-labels and quality-aware weighting. How sensitive are the reported gains to key choices (e.g., partition granularity, flow horizon, weighting thresholds/temperatures) and to moderate segmentation/depth noise?

Beyond the driving benchmarks, how well does the method carry over to out-of-domain conditions (e.g., low-texture scenes, adverse weather, or non-driving datasets)? Are there conditions under which the advantages diminish markedly?

---

### Official Review · Reviewer_XWn3 · 2025-10-27

**Soundness:** 3
**Presentation:** 2
**Contribution:** 3
**Rating:** 4
**Confidence:** 3

**Summary:**

This paper proposed a monocular scene flow estimation pipeline using SAM and DAM as priors. The generation os pesudo labels are evaluated and adjusted by designed loss functions. Extensive experiments prove the competitive performance of proposed methods.

**Strengths:**

1. The injection of depth (scene structure) and semantic information is reasonable, offering better 3D motion learning ability.

2. Authors clearly illustrate the method details and novelty.

3. The overall writing is good and easy to follow.

4. Experiment results are sufficient to prove the effectiveness of the proposed design.

**Weaknesses:**

1. Leveraging foundation models as prior is not so novel. There are some existing methods in optical flow approaches, which utilize foundation models to strengthen the performance. These works share very similar insight.

2. Since VGGT can already predict accurate 3D scene information (point tracking, depth) from monocular images, what is the meaning of designing so many detailed tricks for a better 3D motion learning?

3. In Section 3.1.2, why the input of the network has optical flow? How is it obtained?

4. An evaluation about runtime and efficiency of this proposed method is expected.

5. There are some grammar issues in the paper writing. For example, the inconsistent tenses in the first graph of Section 3.

**Questions:**

Please refer to the weakness sections. Authors should give more motivations why they design the complicated information injection method from SAM and DAM, rather than using VGGT.

---

### Official Review · Reviewer_s2uE · 2025-10-31

**Soundness:** 2
**Presentation:** 2
**Contribution:** 2
**Rating:** 4
**Confidence:** 4

**Summary:**

This paper proposes M3DOnline, a monocular self-supervised 3D scene flow estimation framework trained with foundation (namely, SAM and DAM) models' assistance. This framework applies foundation model outputs to generate pseudo-labels, used to train the model. A pseudo-label pipeline is designed to rectify current estimates based on texture and occlusion regions. A credibility score is also designed to measure the accuracy of pseudo-labels, so they can be used with different weights in training. Experiments show promising results for this method.

**Strengths:**

1. The motivation of the paper is clearly stated, and the proposed method generally makes sense.
2. The figures are well-crafted, which helps understanding.
3. The authors state that the code will be available, representing good reproducibility.

**Weaknesses:**

1. The presentation of the paper needs to be improved, including more formal academic writing for better readability and clarity. Some examples include but are not limited to:
    - Many of the citations in the paper should be cited in the `\citep` format, but the authors used `\citet`, which interrupts the flow of the text.
    - Many typo or grammatical errors:
      - Line 049: "makes" -> "make".
      - Line 178-180: Why is mask $\mathbf{M}_s$ bolded, but the mask $M_b$ not?
      - Line 241: "warp" -> "warped".
      - Line 367-368: "supervision" -> "supervised", "self-supervision" -> "self-supervised".

2. The method design generally needs more context or explanation. For example, Eq (9)-(15): How is these metrics derived? Why cubed in Eq (9), to the power 0.1 in Eq (10)? Why multiplication in Eq (13), not addition? Which metric is scalar and which is vector/map? Any references? The paper does not show any context or explanation on these designs, and they currently just appear out of nowhere.

3. Line 210-211: "Assuming the camera is stationary relative to the world coordinate system". Is this a general assumption for this whole model? I do not believe this assumption holds for autonomous driving.

4. Experiment results:
   - Most of the previous models in Table 3 did not use scene flow datasets other than KITTI to train, so that may not be a very fair comparison. The paper should at least note that in the caption.
   - Table 4: It would be better to highlight the results of your final model, so we can easily tell apart from other ablation experiments.

**Questions:**

See weaknesses.

---

### Meta-Review · Area_Chair_TvD6 · 2026-01-08

**Summary:**

Main concerns were readability, novelty, weak motivation and limited insight, and limited experimental validation.

s2uE: main concerns could be boiled down to writing, including improvement of writing, adding more context, etc... One concern stands out, that is fairness of comparison. Another concern might have been very important during rebuttal was regarding assumption made by the author, which reviewer believed does not "holds for autonomous driving".

XWn3: questioned novelty and need to design structures over existing VGGT.

7ybE: commented that conceptual novelty is limited.

UFHx: was critical of experimental validation, lack of draft sharing motivation of the work and having structural challenges. Reviewer presented detailed set of questions for rebuttal.

**Reviewer Concerns:**

none.

**Reviewer Scores:**

would not have changed much.

Might be s2uE and UFHx might have resisted accept rank.

---

### Decision · Program_Chairs · 2026-01-26

Reject